# Effects of Housing and Management Systems on the Growth, Immunity, Antioxidation, and Related Physiological and Biochemical Indicators of Donkeys in Cold Weather

**DOI:** 10.3390/ani12182405

**Published:** 2022-09-14

**Authors:** Ruiheng Gao, Lulu Shi, Wenliang Guo, Yuanqing Xu, Xiao Jin, Sumei Yan, Binlin Shi

**Affiliations:** College of Animal Science, Inner Mongolia Agricultural University, Hohhot 010018, China

**Keywords:** cold weather, housing and management systems, growth performance, blood biochemical and hormone index, immune and antioxidative function, donkey

## Abstract

**Simple Summary:**

Animal houses are the main place for animal production in intensive breeding and adopting related housing and management systems can improve the environment in houses. The present experiment was designed and conducted to study the effects of housing and management systems on the growth, immunity, antioxidant, and related biochemical indexes of donkeys in a cold climate. The results showed that adopting windproof facilities can effectively improve growth performance, enhance immune function and antioxidant enzyme activity, and maintain relatively steady biochemical indicators of donkeys.

**Abstract:**

The study was designed with a 2 × 2 factorial experiment to evaluate the effects of growth performance, immune function, antioxidant status, blood biochemical indexes, and hormone levels of donkeys in different housing and management systems in cold weather. Twenty-four male donkeys with similar body weight and age were randomly allocated into four treatment groups that were as follows: a cold-water-drinking group without a windproof facility, a lukewarm-water-drinking group without windproof facilities, a cold-water-drinking group with a windproof facility, and a lukewarm-water-drinking group with a windproof facility. The experiment lasted for 42 days. The results showed that windproof facilities increased average daily gain (ADG) and decreased average daily feed intake (ADFI) and feed-to-gain ratio (F:G) at all time periods (*p* < 0.01) of the experiment. Windproof facilities increased the digestibility of dry matter (DM), crude fat (CF), crude protein (CP), ash, calcium (Ca), and phosphate (P) on day 21 (*p* < 0.01), and increased the digestibility of DM, CF, ash, and P on day 42 (*p* < 0.01). The respiration rate and the skin temperature of the abdomen and legs increased (*p* < 0.05) and rectal temperature tended to increase (*p* = 0.083) by adopting windproof facilities at 07:00; the windproof facilities tended to increase the skin temperature of the ears and abdomen (*p* = 0.081, *p* = 0.091) at 14:00. For the blood parameters, with windproof facilities, the concentrations of total protein (TP), blood urea nitrogen (BUN), triglyceride (TG) and high-density lipoprotein cholesterol (HDL-C) increased (*p* < 0.05) and glucose (GLU) concentration decreased (*p* < 0.05) at 07:00 on day 21; the concentrations of TG and cholesterol (CHO) increased and the concentrations of TP, BUN, and GLU decreased at 07:00 on day 42 (*p* < 0.05). The concentrations of adrenocorticotropic hormone (ACTH), cortisol (COR), triiodothyronine (T3), and thyroxine (T4) decreased (*p* < 0.05) at 07:00 on day 21, and T4 concentration decreased (*p* < 0.05) at 07:00 on day 42. The concentrations of interleukin-4 (IL-4), immunoglobulin A (IgA), immunoglobulin G (IgG), and immunoglobulin M (IgM) increased (*p* < 0.01) and the concentrations of interleukin-1β (IL-1β) and tumor necrosis factor-alpha (TNF-α) decreased (*p* < 0.01) on days 21 and 42. The activities of total superoxide dismutase (T-SOD) and glutathione peroxidase (GPx) increased (*p* < 0.05), and malondialdehyde (MDA) concentration decreased (*p* < 0.01) on day 21; the activities of T-SOD and catalase (CAT) increased (*p* < 0.05), and MDA concentration decreased (*p* < 0.01) on day 42. However, under the conditions of this experiment, water temperature did not affect the above indexes on days 21 and 42. These results indicated that adopting windproof facilities in a cold climate can mitigate the effects of atrocious weather on the production performance of donkeys.

## 1. Introduction

The donkey, Equus africanus asinus, is a domesticated member of the Equidae or horse family. China is one of the major donkey breeding countries in the world with a history of nearly 4000 years. They are the most important draught animals playing a key role in the agriculture sectors in developing countries. They serve as pack animals, and for carting, threshing, farm cultivation, riding, milk, and meat production for humans [1]. However, with the continuous development of social productivity, donkeys have gradually been withdrawn from the traditional servitude work, and have attained increased importance in meat and medicine production. Despite the great economic importance of donkeys in many countries of the world, people rarely use scientific and effective methods to breed donkeys, and donkeys’ welfare is also often neglected. However, it is important to protect them from harsh climatic changes.

Environmental factors can affect the physiological metabolism of animals. Extreme cold weather will change the homeostasis of animals and lead to cold stress [2,3]. Temperature is an important environmental factor. When the ambient temperature drops, the animal will experience cold stress [4]. Cold stress is a common adaptive response. Under the action of cold stress, animals will transfer a large amount of energy to generate body heat, thus sacrificing growth, lactation, reproduction, and other functions [5,6,7]. Cold stress negatively interferes with production by limiting feed conversion efficiency. Even if the animals increase feed intake, this is not sufficient to supply the metabolizable energy required to maintain body temperature, resulting in economic loss [8]. At the same time, animals show physiological and behavioral changes in cold conditions, such as bristling, curling, and trembling. If animals are exposed to a cold environment for a long time, the secretion of thyroid hormones, adrenal cortex hormones, and other hormones will increase [9], and the related physiological metabolic activities will also change [10]. Cold stress influences many cellular processes that in turn lead to physiological and immunological responses [11]. Cells can develop efficient stress responses by activating protein control systems (i.e., gene transcription, protein expression, and enzyme activity) or proceed into cell-death signaling pathways to cope with the environment [12]. Cold stress can increase endogenous reactive oxygen species (ROS) [13,14]. ROS generation may be a requirement for DNA damage and apoptotic events. In a normal functioning and healthy immune system, apoptosis is used to regulate the number of immune cells before and after the immune response [15].

Along with the development of feeding technologies, it is necessary to create a suitable microclimate in animal houses, to allow animals to realize their productive potential. However, intensive breeding and high animal density increase the microclimate requirements in buildings. The formation of the microclimate in animal houses depends on a number of factors—the natural climatic conditions, the type and quality of the buildings, the type of construction materials, and the ways of raising the animals, their density. Natural climatic factors that influence the formation of microclimates in animal feeding sites include the temperature, the intensity, and duration of solar radiation. Natural climatic characteristics unique to individual areas should be considered when building new houses and restoring old ones. There has been relatively little research on the impact of climate change on livestock production compared to crop production. Escarcha et al. [16] pointed out in their systematic literature review that only 6% of the research analyzed the impact of climate change on livestock husbandry. Lack of quantitative research on livestock and poultry construction makes it difficult for livestock and poultry managers to select housing and management systems. Limited research has been conducted on the effects of housing and management systems on growth performance and physiological function in donkeys. In this sense, the present study aimed to examine the effects of different housing and management systems in cold weather on growth performance, immune function, antioxidant status, and blood biochemical and hormone levels of donkeys.

## 2. Materials and Methods

### 2.1. Animals and Experiment Design

The experiment was conducted in the Inner Mongolia Caoyuanyulv Science and Technology Animal Husbandry Co., Ltd., Hohhot, Inner Mongolia, China, from 9 December, 2019, to 19 January, 2020, and lasted for 42 days. The study was designed as a 2 × 2 factorial experiment. A total of 24 male donkeys with similar body weight and age (BW = 128 ± 8 kg, age = 6 months) were randomly allocated into 4 equal treatment groups. The experiment was carried out in the same semi-open barn. The barn as a whole was bell tower type (double sloped roof with double side skylights in the middle of the roof), and there were many separate fences built in the barn for raising animals. The south wall of the barn was only half the height of the rest (1.70 m) and the sealing was very poor. The natural environment had a great impact on the environment in the barn. We divided the barn on average into the east and west parts. Transparent plastic PVC film (0.12 mm thick) was used to cover the open part in the east part of the barn so as to reduce the impact of the natural environment on the barn’s environment. The west part of the barn remained untreated. We divided the donkeys fed in the east half of the barn into two groups; one group drank cold water (0~5 °C), the other group drank lukewarm water (25~30 °C), and the same was true for the donkeys fed in the west part of the barn. There were doors on both sides of the barn through which the experimenters entered the barn for providing feed and water. Lukewarm water was collected from the water heater and quickly supplied to the donkeys to ensure water temperature. All treatment groups were fed the same total mixed ration at 7:00, and cereal grass at 10:00, 13:00, 17:00, and 20:00. The water was drunk at 11:00–11:30 and 17:00–17:30 every day during the experiment.

All animal experiments were performed in accordance with the national standard Guideline for Ethical Review of Animal Welfare (GB/T 35892-2018).

### 2.2. Meteorological Observation

Data of air temperature and relative humidity in the donkey stalls were recorded daily with thermohygrometers, and the wind speed inside and outside the stall at 08:00, 14:00, and 20:00 with a hot-bulb anemometer during the period of the experiment.

### 2.3. Measurement of Growth Performance and Nutrient Digestibility

The donkeys were individually weighed using a scale (XK3190-A12E, Taimai Weighing Instrument Co., Ltd., Shanghai, China) at the start of the experiment and on days 21 and 42. Feed consumption was recorded daily and the average daily gain (ADG), average daily feed intake (ADFI), and feed-to-gain ratio (F:G) were calculated.

Daily fecal samples and feed samples for each donkey were collected, weighed, and mixed thoroughly for five consecutive days from days 17 to 21 and days 38 to 42. The collected samples were dried in a forced draft oven at 65 °C for at least 48 h until constant weight, and then ground through a 1-mm screen in a mill. Dry matter (DM) concentration was assessed by oven drying (DGX-9053B, Fuma Experimental Equipment Co., Ltd., Shanghai, China) at 105 °C for 8 h, and ash concentration was assessed by a muffle furnace (XL-I, Hengbo Instrument & Meter Co., Ltd., Hebi, China) at 550 °C for 6 h, as described in AOAC (2004) [17]. The fecal and feed samples were analyzed for crude fat (CF), crude protein (CP), calcium (Ca), phosphate (P), neutral detergent fiber (NDF), and acid detergent fiber (ADF) according to AOAC (2004) [17]. The CF concentration was analyzed using Soxhlet extraction (SOX406, Haineng Scientific Instrument Co., Ltd., Shandong, China), and the CP concentration was determined using the Kjeldahl method (K9840, Haineng Scientific Instrument Co., Ltd., Shandong, China) of nitrogen analysis. The NDF and ADF concentration was analyzed using the filter bag technique (ANKOM A200i, ANKOM Technology, New York, NY, USA). Acid insoluble ash (AIA) content was used as an internal marker to determine the apparent digestibility of the experimental diets [18].

### 2.4. Measurement of Basic Physiological Indexes

Respiration rate, skin temperature (head, ear, abdomen, and leg), and rectal temperature were measured at 07:00 and 14:00 on days 7, 14, 21, 28, 35, and 42. Respiration rate: the thoracic and abdominal movements of the donkey were observed and recorded within 1 min when the donkey was quiet. Skin temperature: an infrared thermometer was used to measure at a distance of about 2 cm from the measured site (forehead, behind the right ear, abdomen, and inside the right hind leg); Rectal temperature: the rectal temperature of the donkey was measured with an animal rectal thermometer, and the thermometer was disinfected after each donkey measurement.

### 2.5. Preparation and Analysis of Blood Sample

The jugular vein blood samples were collected at 07:00 and 14:00 on days 21 and 42. Blood samples were centrifuged for 15 min at 3500× *g* to collect serum and then frozen at −20 °C. Blood biochemical indicators included total protein (TP), albumin (ALB), triglyceride (TG), cholesterol (CHO), blood urea nitrogen (BUN), glucose (GLU), high-density lipoprotein cholesterol (HDL-C), and low-density lipoprotein cholesterol (LDL-C), which were determined by an automatic biochemistry analyzer HITACHI 7020 (Japan) according to the instrument’s instructions. Hormone and immune parameters, including the concentrations of adrenocorticotropic hormone (ACTH), cortisol (COR), adiponectin (ADPN), insulin (INS), leptin (LEP), triiodothyronine (T3), thyroxine (T4), interleukin-1β (IL-1β), interleukin-4 (IL-4), tumor necrosis factor-alpha (TNF-α), immunoglobulin A (IgA), immunoglobulin G (IgG), and immunoglobulin M (IgM), were determined with donkey-specific ELISA kits (Ruixin Biological Technology Co., Ltd., Quanzhou, China), according to the manufacturer’s instructions. Antioxidative parameters, including the activities of glutathione peroxidase (GPx), total superoxide dismutase (T-SOD), catalase (CAT), malondialdehyde (MDA) concentration, and total antioxidant capacity (T-AOC) were measured using commercial kits according to the instructions of the manufacturer (Nanjing Jiancheng Institute of Bioengineering, Nanjing, China).

### 2.6. Statistical Analysis

Data were tested for the main effects of wind and water temperature and their interactions using the GLM procedure of SAS version 9.2 (SAS Institute Inc., Cary, NC, USA). Taking windproof facilities and drinking water temperature as the main influencing factors, the influence of and interaction between windproof facilities and drinking water temperature were analyzed. The multiple comparisons among groups were done using Duncan’s test. Probability values of *p* < 0.05 were used as the criterion for statistical significance, and *p* < 0.10 was considered a tendency. Data in tables were presented as the means and standard error of the mean (SEM), and data in figures were presented as mean ± SD.

## 3. Results

### 3.1. Ambient Temperature and Wind Speed

As can be seen from Figure 1, the ambient temperature of different groups and the outdoor temperature had the same change trend, reaching the maximum daily temperature at 14:00, but there were differences in the ambient temperature of different treatment groups. The temperature in the barn with windproof facilities was higher than that in the barn without windproof cover all day, and the maximum temperature difference reached 3.9 °C. The mean wind speeds of different groups at 08:00, 14:00, and 20:00 are presented in Figure 2. Although there was no obvious variance tendency of wind speed at different times, the wind speed of the barn with windproof facilities was always lower than that in the barn without windproof facilities, and the wind speed of the barn with windproof facilities was very low, close to the state of no wind.

### 3.2. Growth Performance and Nutrient Digestibility

The effects of housing and management systems on the growth performance of donkeys are presented in Table 1. Adopting windproof facilities increased ADG and decreased ADFI and F:G during all time periods (*p* < 0.01) of the experiment; however, the water temperature had no significant influence on ADG, ADFI, and F:G. Comparing the water intake, it could be found that the intake of lukewarm water was greater than that of cold water in the barn without windproof facilities (*p* < 0.01), but there was no significant difference between lukewarm water intake and cold water intake in the barn with windproof facilities.

The effects of housing and management systems on the nutrient digestibility of donkeys are presented in Table 2. Adopting windproof facilities increased the digestibility of DM, CF, CP, Ash, Ca, and P on day 21 (*p* < 0.01), and increased the digestibility of DM, CF, Ash, and P on day 42 (*p* < 0.01). In addition, the digestibility of all nutrients was not affected by water temperature on days 21 and 42.

### 3.3. Basic Physiological Indexes

The effects of housing and management systems on the basic physiological indexes of donkeys are presented in Table 3. When measured on day 21, adopting windproof facilities increased the skin temperature of legs and rectal temperature (*p* < 0.05) at 07:00, increased the skin temperature of heads and abdomen (*p* < 0.05) at 14:00, and tended to increase the skin temperature of heads (*p* = 0.091) and ears (*p* = 0.078) at 7:00 and rectal temperature (*p* = 0.093) at 14:00. When measured on day 42, adopting windproof facilities increased the skin temperature of heads (*p* < 0.05) at 07:00 and 14:00, increased the rectal temperature (*p* < 0.05) at 14:00, and tended to increase the skin temperature of legs (*p* = 0.059, *p* = 0.077) at 7:00 and 14:00 and respiratory rate (*p* = 0.055) at 14:00. Windproof facilities play a major role in the changes in basic physiological indexes.

### 3.4. Blood Biochemical and Hormone Indexes

The effects of housing and management systems on the biochemical indicators of donkeys are presented in Table 4. Adopting windproof facilities increased the content of TP, BUN, TG, and HDL-C (*p* < 0.05) and decreased GLU content (*p* < 0.05) at 07:00 on day 21; meanwhile, it increased the content of TG and CHO (*p* < 0.05) and decreased the content of TP, BUN, and GLU (*p* < 0.05) at 07:00 on day 42, but had no significant effect on blood biochemical indexes at 14:00 on days 21 and 42. Similarly, the water temperature had no impact on any of the biochemical indexes.

The effects of housing and management systems on the hormone levels of donkeys are presented in Table 5. Adopting windproof facilities decreased the content of ACTH, COR, T3, and T4 at 07:00 on day 21 (*p* < 0.05), decreased T4 content at 07:00 on day 42 (*p* < 0.05), and had no impact on other hormone indexes. In addition, the water temperature had no impact on hormone levels.

### 3.5. Immune and Antioxidative Indexes

The effects of housing and management systems on the immune indexes of donkeys are presented in Table 6. Adopting windproof facilities increased IL-4 content (*p* < 0.01) and decreased the content of IL-1β and TNF-α (*p* < 0.01) on day 21. In the barn without windproof facilities, drinking lukewarm water decreased IL-1β content (*p* < 0.01) on day 21. In addition, adopting windproof facilities increased the content of IgA, IgG, and IgM on day 21 (*p* < 0.01), but water temperature had no impact on immunoglobulin content. On day 42, the effect of windproof facilities on immune indexes was the same as that on day 21.

The effects of housing and management systems on the antioxidative indexes of donkeys are presented in Table 7. Adopting windproof facilities increased the activity of T-SOD and GPx (*p* < 0.05), and decreased MDA content (*p* < 0.01) on day 21. On day 42, adopting windproof facilities increased the activity of T-SOD and CAT (*p* < 0.05), and decreased MDA content (*p* < 0.01). However, the water temperature had no impact on antioxidative indexes.

## 4. Discussions

Although donkeys are homothermal animals that can adopt adaptive mechanisms to address cold stress and re-establish the dynamic balance of body temperature, sudden severe cold stress may be harmful to donkeys [19]. Cold stress increased basal metabolic rate and energy metabolism, and endotherms have higher priority and energy requirements [20]. The increase in energy demand for generating heat is the main reason for the decrease in BW gain [21]. It has been reported that cold stress decreased the feed conversion rate and reduced the BW gain of calves [22]. Our findings showed that windproof facilities increased average daily gain and decreased average daily feed intake, and also suggested that cold stress decreased feed conversion ratio, which is consistent with previous studies, suggesting the redistribution of nutrients from growth toward thermoregulatory responses. Kennedy et al. [23] conducted cold stress tests on sheep and found that when the ambient temperature decreased by 1 °C, the feed digestibility of sheep decreased by 0.31%. Graham et al. [24] found that when the temperature decreased by 10 °C from 25 °C, the dry matter digestibility of sheep decreased by 12%. Nicholson et al. [25] pointed out that in a certain range of low temperatures, the nutrient digestibility of sheep was positively correlated with environmental temperature, and the digestibility of dry matter of sheep decreased with the decrease in temperature. Compared with open barns, closed barns increased the nutrient digestibility of cows [26]. There is speculation that the speed at which food moves through the digestive tract may be one reason for the reduced digestibility of feed when animals are exposed to cold conditions. There is also a view that when animals experience cold stress, the impact on the digestive system may cause stress ulcers, and the occurrence of ulcers in the stomach is relatively high in rats [27]. When ulcers occur, the efficiency of animal digestion of food will be reduced.

The main response of animals to cold conditions is to increase their own heat production and reduce their heat loss, thus maintaining their body temperature within the normal range. Like large animals, donkeys have a lower ratio of body surface area to body weight, thicker skin, and subcutaneous fat layer, which allow them to resist cold stress more effectively than small animals [22]. However, long-term exposure to a cold environment still has a certain influence on physiological indicators. Sykes and Slee [28] reported that the skin temperature of Scottish blackface sheep under either acute cold stress or chronic cold stress was lower than ambient temperatures, and the reduction in skin temperature reduced the difference between the skin temperature and the ambient temperature, thus reducing the body heat dissipation. Diesel et al. [29] pointed out that when animals are subjected to low-temperature stress, the body reduces temperature loss and maintains normal temperature by regulating the respiratory system to deepen and slow respiration. Quickening and deepening of respiration are cold-resistant responses to the increase in metabolism and heat production of the body. With the increase in metabolic rate, the oxygen demand of the body will also increase, resulting in accelerated respiration of the body. These results may indicate that in a cold environment, in order to maintain the stability of the internal environment, the body produces more heat and reduces heat dissipation. However, when the ambient temperature exceeds the body’s own regulation limit, the respiration deepens and the frequency decreases, thus reducing the heat dissipation of the respiratory tract.

Seasonal and environmental changes can affect hematological values in domestic animals [30]. Hematological parameters have been used to identify the effect of heat and cold stress on productivity in animals. As a physiological response index, blood parameter spectrum can be used as the basis for diagnosis, treatment, and prognosis of diseases [31]. Hormones, including glucocorticoids and thyroid hormones, might regulate immune function in response to environmental stimuli, such as physical and emotional stress, nutritional deprivation, and environmental temperature [32]. Therefore, biochemical indexes in animal blood can reflect changes in animal metabolic function. Cold exposure is one of the typical patterns of energy challenge, and a common response to this challenge is the mobilization of stored fat by adipose tissues to maintain the critical functions of different tissues [33]. LDL-C is the main tool for transporting cholesterol to extrahepatic tissues and the most important form of cholesterol in the blood. Cold stress can change the concentration of serum HDL-C, LDL-C, and other cholesterol components, indicating that cold stress can accelerate fat dissolution and provide energy for the body. Consistent with the results of Yu et al. [34] and Zhang et al. [35], our results revealed that windproof facilities decreased the level of fat metabolism in donkeys, the energy required for heat production is reduced, and more energy is provided for growth performance, alleviating the effect of cold stress on donkeys. Serum protein metabolism level can be reflected by TP, ALB, and GLB content. BUN is the downstream end product of metabolic reactions of animal protein and amino acid, and its content can reflect the balance of protein metabolism. Generally speaking, serum urea nitrogen content is positively correlated with protein content, and young animals are more sensitive to changes in protein content due to the imperfect development of renal metabolic function [36]. In this study, protein metabolism also showed changes similar to those in fat metabolism, but protein metabolism showed opposite changes at the later stage of the experiment, which might be the compensatory response of donkeys exposed to long-term cold stress. It has been reported that glucagon mediates the liver to break down stored glycogen into GLU during cold stress or short periods of fasting; GLU is then released into the circulation to maintain GLU homeostasis in the blood during periods of high energy demand or fasting [37]. Young [6] suggested that increased blood glucose results from increased metabolism (e.g., metabolic rate or heart rate) under cold conditions. In the present study, we found that cold stress increased serum GLU in donkeys. This result might be due to the stress-induced activation of the hypothalamic–pituitary–adrenal axis. An increase in cortisol leads to an increase in blood GLU levels, which can help animals cope with environmental threats [38]. It is worth noting that, compared with previous studies, the results of the present study were within the reference ranges at all time points [39]. Nonetheless, the differences between the four treatment groups suggest that different housing and management systems can affect energy allocation in cold weather, and thus affect donkey growth performance.

Cold stress has become an important factor that restricts the development of animal husbandry in the cold region [40]. Hormones, as important substances in the blood regulating the body’s normal activities, can promote the normal growth and development of tissues and organs, and are regulated by the neuroendocrine system to enable the body to adapt to changes in the environment. Hypothalamic–pituitary–adrenal (HPA) activity is thought to be an adaptive response during repeated exposure to the same stressor, so the ACTH and COR responses of chronically stressed animals to the same stressor are lower than those of animals first exposed to the same stressor [41]. In contrast, the presence of a new or heterotypic stressor leads to elevated levels of ACTH and COR in the plasma of chronically stressed animals, and much higher levels in the plasma of control animals [42]. Similarly, adaptation to the homotypic stressor and hyperresponsiveness to a heterotypic stressor do not occur in all chronic stress paradigms [43] but are instead determined by the nature of the stressors [44]. As for the continuous influence of chronic stress on HPA, relevant reports indicate that the basal HPA activity may or may not be increased following exposure to chronic intermittent stress [45]. The different change patterns of ACTH and COR in the early and late periods of this study also demonstrated this view. In addition, there are some reports that plasma ACTH responses to cold stress are transient, and that plasma levels of glucocorticoid or thyroid-stimulating hormone (TSH) are not changed during cold exposure [9]. These discrepancies may be caused by different experimental conditions, such as the intensity of cold exposure, sampling time, and procedures.

Thyroid hormones mainly include T3 and T4, and T4 is the main secretion of the thyroid gland while most of T3 is transformed by deiodination of T4, which can enhance tissue metabolism, increase tissue heat production, and promote carbohydrate absorption. Fukuhara et al. [9] reported that both 1- and 5-day hypothermia exposure increased plasma levels of TSH and thyroid hormones, suggesting that this stressor chronically activated the hypothalamic–pituitary–thyroid (HPT) system in rats. McBride et al. [19] reported that short-term cold exposure had no effect on plasma T3 and T4 in adult horses, but period cold exposure significantly increased plasma T3 and T4 in adult horses. In the present study, T3 and T4 levels showed the same results. This result is consistent with the view that maintaining body temperature in a cold environment requires an increase in T3, which is highly active as a heat producer. A comparison of serum T3 and T4 levels in different periods showed that, in the early period, they were higher than those in the late period, which indicated that prolonged cold stress results in acclimation and restoration of heat balance, as can be inferred from the normalized serum T3 levels as well as progressively decreased serum ACTH levels. Since glucocorticoid hormones initially potentiate to secrete, and later suppress the cold-induced release of TSH, it appears that chronic activation of the HPA system in cold-stressed animals is important for maintaining the secretion of TSH.

The response of the immune system is one of the mechanisms by which organisms resist environmental challenges. Immune response and inflammatory response are important physiological processes in regulating immune function. Acute and chronic cold stress often have different effects on immune response, and chronic cold stress most often leads to suppression of the immune system. However, acute cold stress has limited suppressive effects on immune function. The most common theory is that cold stress suppresses components of the immune system, thus increasing animals’ susceptibility to diseases. In addition, some studies have shown that cold stress enhances the immune system, while others have shown that cold stress has no effect on immunity, mainly because of the complexity of the immune and stress systems, and other factors (i.e., age, genetics, physiological status) that influence the stress response of animals [46].

T helper cells (Th) could be divided into two groups according to different patterns of cytokine secretion, known as Th1 and Th2, which play a crucial role in cellular and humoral immunity through self-regulation and co-adjustment [47,48]. Th1 mainly secretes IL-1 and TNF-α, while Th2 mainly secretes IL-4 and IL-6 [49]. Maintaining Th1/Th2 balance is of great significance to sustaining normal immune function [50]. Therefore, under homeostatic conditions, Th1 and Th2 responses normally exist in balance and keep each other in check. For most infections, except those caused by metazoan parasites, Th1 responses are protective, while Th2 responses assist in the resolution of cell-mediated inflammation. TNF-α and IL-1, as the main pro-inflammatory factors in the early inflammatory response, can bind with a variety of receptors on immune cells and play an immunomodulatory role. However, with the increase in their secretion, the body’s inflammatory response will intensify, and eventually cause tissue damage and reduce the body’s immune function [51]. IL-4 is a multifunctional cytokine secreted by Th2 cells, mainly used as an anti-inflammatory cytokine to regulate inflammation. It can promote the growth and differentiation of Th2 cells and induce B cells to secrete immunoglobulins, thereby increasing the level of antibodies [52]. In the present study, adopting windproof facilities decreased the production of serum IL-1β and TNF-α, and increased the production of serum IL-4 in donkeys. Similar to the current findings, Cong et al. [53] demonstrated that cold stress could cause immune cell damage, greatly reduce the number of white blood cells and monocytes in the blood, and thus inhibit the secretion of immune cytokines. Immunoglobulin is a kind of protein with antibody activity, which mainly exists in plasma, and is used to promote the phagocytosis of monocytes and macrophages, identify and neutralize bacteria and viruses, and plays an important role in specific immune response [54]. Serum IgG is the most persistent and important antibody in the primary immune response, and is used to promote phagocytosis of monocytes and macrophages, neutralize the toxicity of bacterial toxins, and combine with viral antigens to prevent the virus infecting host cells [55]. Our findings showed that serum levels of IgA, IgG, and IgM were increased in donkeys housed in windproof facilities, indicating that windproof facilities efficiently modulated a humoral immune response, which might provide an advantage in increasing immunity to decrease susceptibility to disease and enhance resistance to infections in donkeys.

Oxidative stress can damage biomolecules, including cellular lipids, proteins, amino acids, and deoxyribonucleic acid, thereby inhibiting their normal functions [56,57]. In addition, when organisms are stimulated by endogenous and exogenous stimuli, partially reduced and highly reactive O_2_ metabolites are formed to produce superoxide anions (O_2^−^_) and hydrogen peroxide (H_2_O_2_). These derivatives of oxygen (reactive oxygen species, e.g., O_2^−^_ and H_2_O_2_) are highly reactive and toxic, and can lead to increased oxidative stress in a variety of tissues [58]. Malondialdehyde is the most common decomposition product of lipid peroxidation, and its level is often used as a major marker to measure the degree of lipid oxidative damage caused by ROS. This study’s results revealed that windproof facilities can reduce the serum MDA content in donkeys, suggesting that windproof facilities can depress their lipid peroxidation. Antioxidant enzymes are key factors in the animal defense system against stimulus-induced oxidative stress. The enzyme defense system consists of SOD, CAT, and GPx. SOD provides the efficient dismutation of superoxide radicals (O_2^−^_) into less toxic H_2_O_2_, while CAT and GPx degrade H_2_O_2_ to O_2_ and H_2_O [59]. Gumuslu et al. [60] reported that a cold environment decreased the activities of SOD, CAT, and GPx in rat erythrocytes. Additionally, Sahin and Gumuslu [61] concluded that cold stress decreased hepatic, nephritic, and cardiac SOD, CAT, and GPx activities in rats. In agreement with these findings, the present results showed that windproof facilities increased serum SOD, CAT, and GPx activities. Previous research results have shown that cold stress can cause oxidative damage to serum, tissues, and organs to different degrees.

## 5. Conclusions

In conclusion, the growth performance, immune function, and antioxidant status of donkeys can be improved by adopting windproof facilities in a cold climate, and the health status, blood biochemical, and hormone levels of donkeys can be maintained within the normal range. However, under the conditions of this research, the temperature of drinking water had no effect on the physiological state and functioning of donkeys. Therefore, it can be suggested that adopting windproof facilities in a cold climate can mitigate the effects of atrocious weather on the production performance of donkeys.

## Figures and Tables

**Figure 1 animals-12-02405-f001:**
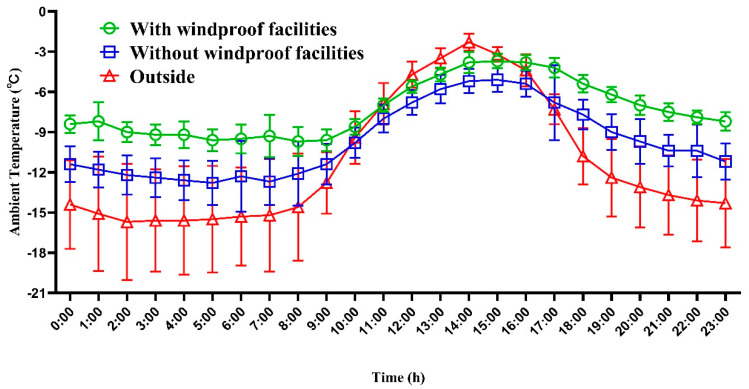
Daily variation of temperature in the donkey house. During the test period (42 days), the hourly temperature was selected for statistics, and the results were expressed as mean ± SD. The temperature of the drinking water had no effect on the temperature of the barn, so only the ambient temperature of the right part of the barn (with windproof facilities), the left part of the barn (without windproof facilities), and the outside were counted.

**Figure 2 animals-12-02405-f002:**
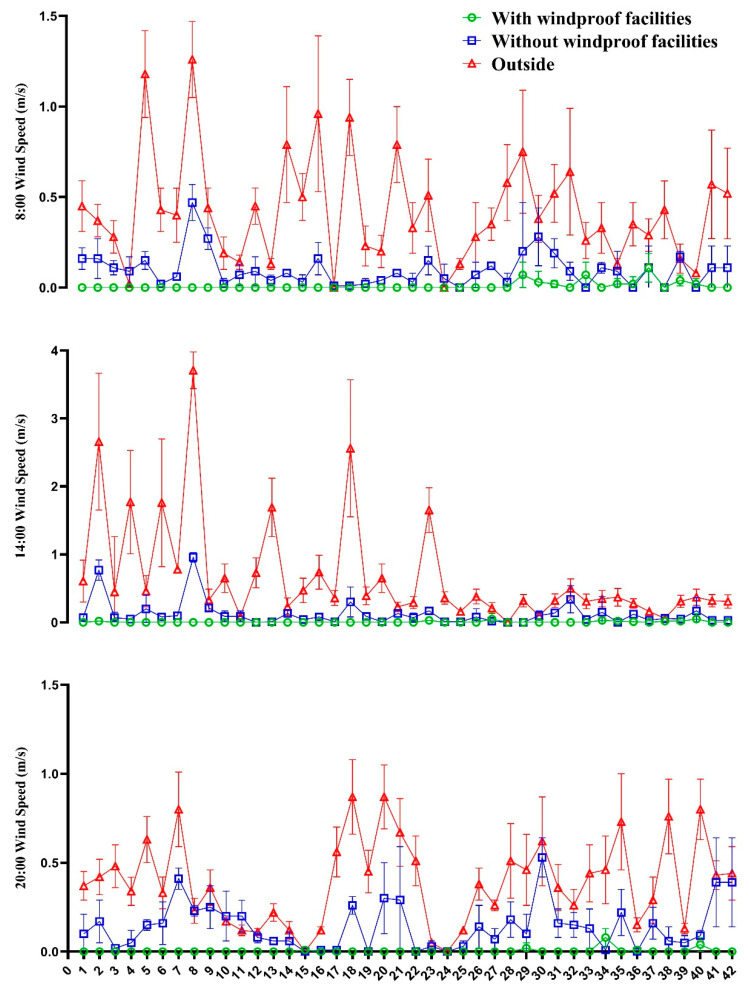
Change in wind speed in the donkey house. During the test period (42 days), the wind speed at 8:00, 14:00, and 20:00 was selected for statistics, and the results were expressed as mean ± SD. The temperature of the drinking water had no effect on the temperature of the barn, so only the ambient temperature of the right part of the barn (with windproof facilities), the left part of the barn (without windproof facilities), and the outside were counted.

**Table 1 animals-12-02405-t001:** Effects of housing and management systems on the growth performance of donkeys in cold weather.

Items	No Windproof f.		Windproof f.		SEM	*p*	*p*		
	Cold Water	Lukewarm Water	Cold Water	Lukewarm Water			WF	WT	WF × WT
IBW kg	127.6	128.8	128.0	127.9	1.80	0.996	-	-	-
ADG kg/d									
1–21 d	0.241 ^b^	0.244 ^b^	0.299 ^a^	0.309 ^a^	1.89	<0.001	<0.001	0.132	0.359
22–42 d	0.254 ^b^	0.266 ^b^	0.348 ^a^	0.333 ^a^	2.61	<0.001	<0.001	0.821	0.021
1–42 d	0.248 ^b^	0.255 ^b^	0.323 ^a^	0.321 ^a^	1.90	<0.001	<0.001	0.538	0.224
ADFI g/d									
1–21 d	4029 ^ab^	4270 ^a^	3685 ^c^	3955 ^bc^	50.0	0.005	0.004	0.019	0.888
22–42 d	4230 ^ab^	4453 ^a^	4061 ^b^	4042 ^b^	46.9	0.020	0.006	0.290	0.214
1–42 d	4129 ^ab^	4361 ^a^	3873 ^b^	3998 ^b^	47.1	0.010	0.004	0.073	0.577
F:G									
1–21 d	17.32 ^a^	16.90 ^a^	12.58 ^b^	12.57 ^b^	0.20	<0.001	<0.001	0.597	0.609
22–42 d	17.43 ^a^	16.01 ^b^	11.69 ^c^	12.16 ^c^	0.17	<0.001	<0.001	0.177	0.011
1–42 d	17.38 ^a^	16.42 ^a^	12.10 ^b^	12.36 ^b^	0.17	<0.001	<0.001	0.328	0.094
Water intake L/d									
1–21 d	3.68 ^c^	4.93 ^a^	4.21 ^bc^	4.61 ^ab^	0.11	0.005	0.645	0.002	0.073
22–42 d	3.79 ^b^	4.82 ^a^	4.06 ^b^	4.29 ^ab^	0.11	0.019	0.552	0.008	0.073
1–42 d	3.73 ^c^	4.87 ^a^	4.12 ^bc^	4.44 ^ab^	0.10	0.006	0.923	0.002	0.059

^a, b, c^ Different superscripts within each row indicate significant differences (*p* < 0.05). IBW: initial body weight; ADG: average daily gain; ADFI: average daily feed intake; F:G: feed-to-gain ratio; WF: windproof facilities; WT: water temperature; WF × WT: the interaction between windproof facilities and water temperature.

**Table 2 animals-12-02405-t002:** Effects of housing and management systems on the nutrient digestibility of donkeys in cold weather.

Items	No Windproof f.		Windproof f.		SEM	*p*	*p*		
	Cold Water	Lukewarm Water	Cold Water	Lukewarm Water			WF	WT	WF × WT
21 d									
DM	72.20 ^b^	72.17 ^b^	73.88 ^a^	74.01 ^a^	0.18	0.004	0.005	0.064	0.053
CF	60.52 ^b^	62.26 ^ab^	64.38 ^ab^	66.91 ^a^	0.65	0.013	0.007	0.840	0.276
CP	63.99 ^b^	65.74 ^ab^	67.23 ^a^	67.44 ^a^	0.32	0.003	0.140	0.237	0.001
Ash	29.56 ^b^	27.02 ^b^	35.97 ^a^	36.53 ^a^	0.50	<0.001	0.164	0.079	0.384
Ca	57.08 ^bc^	54.15 ^c^	59.05 ^ab^	61.10 ^a^	0.61	0.005	0.361	0.177	0.957
P	28.35 ^b^	26.88 ^b^	32.51 ^a^	32.94 ^a^	0.40	<0.001	<0.001	0.337	0.140
NDF	54.97	53.28	52.63	51.24	0.56	0.345	0.002	0.719	0.054
ADF	35.31	34.61	34.24	33.45	0.38	0.455	<0.001	0.532	0.255
42 d									
DM	66.19 ^b^	66.55 ^b^	69.26 ^a^	70.14 ^a^	0.20	<0.001	<0.001	0.132	0.523
CF	65.02 ^b^	64.56 ^b^	71.87 ^a^	71.94 ^a^	0.94	0.011	0.001	0.920	0.891
CP	61.72	61.84	61.62	61.04	0.31	0.808	0.479	0.719	0.578
Ash	29.82 ^b^	31.19 ^b^	36.54 ^a^	38.92 ^a^	0.52	<0.001	0.700	0.388	0.841
Ca	49.45	52.23	52.96	52.24	0.66	0.277	0.901	0.097	0.994
P	27.43 ^b^	27.41 ^b^	31.81 ^a^	31.94 ^a^	0.32	<0.001	<0.001	0.084	0.633
NDF	49.01	48.20	47.53	47.28	0.48	0.807	0.198	0.442	0.200
ADF	34.16	34.03	32.19	32.04	0.39	0.406	<0.001	0.928	0.909

^a, b^ Different superscripts within each row indicate significant differences (*p* < 0.05). DM: dry matter; CF: crude fat; CP: crude protein; Ca: calcium; P: phosphate; NDF: neutral detergent fiber; ADF: acid detergent fiber; WF: windproof facilities; WT: water temperature; WF × WT: the interaction between windproof facilities and water temperature.

**Table 3 animals-12-02405-t003:** Effects of housing and management systems on the basic physiological indexes of donkeys in cold weather.

Items	Time	No Windproof f.		Windproof f.		SEM	*p*	*p*		
		Cold Water	Lukewarm Water	Cold Water	Lukewarm Water			WF	WT	WF × WT
21 d										
Head °C	7:00	8.96 ^b^	9.73 ^ab^	10.30 ^ab^	10.78 ^a^	0.54	0.091	0.039	0.261	0.796
	14:00	9.07 ^b^	8.93 ^b^	10.30 ^ab^	11.72 ^a^	0.48	0.020	0.001	0.199	0.125
Ear °C	7:00	7.60 ^b^	8.92 ^a^	8.80 ^ab^	8.65 ^ab^	0.40	0.078	0.163	0.260	0.084
	14:00	9.58	10.12	10.92	11.07	0.47	0.125	0.026	0.691	0.480
Abdomen °C	7:00	12.87	12.70	11.83	12.70	0.34	0.171	0.149	0.322	0.149
	14:00	10.97 ^b^	11.60 ^b^	13.03 ^a^	13.52 ^a^	0.47	0.003	0.004	0.252	0.876
Leg °C	7:00	11.92 ^b^	12.68 ^b^	14.52 ^a^	15.58 ^a^	0.46	<0.001	<0.001	0.542	0.032
	14:00	13.75	13.10	13.55	14.28	0.61	0.580	0.426	0.946	0.267
Rectal temperature °C	7:00	36.43 ^b^	36.45 ^b^	36.85 ^a^	36.92 ^a^	0.13	0.021	0.003	0.847	0.747
	14:00	37.35 ^b^	37.38 ^b^	37.45 ^ab^	37.75 ^a^	0.12	0.093	0.062	0.174	0.272
Respiratory rate	7:00	10.85	11.50	11.45	11.38	0.49	0.772	0.560	0.628	0.475
	14:00	12.46	12.00	11.60	13.28	0.54	0.189	0.286	0.681	0.070
42 d										
Head °C	7:00	10.37 ^b^	10.83 ^b^	11.23 ^b^	12.47 ^a^	0.38	0.005	0.003	0.035	0.321
	14:00	9.63 ^b^	9.68 ^b^	10.10 ^ab^	10.87 ^a^	0.31	0.038	0.201	0.260	0.015
Ear °C	7:00	9.23	9.50	9.65	10.37	0.52	0.444	0.216	0.339	0.659
	14:00	9.88	10.50	10.68	10.85	0.41	0.394	0.179	0.354	0.592
Abdomen °C	7:00	10.17	11.03	10.85	11.20	0.40	0.301	0.141	0.298	0.523
	14:00	11.70 ^b^	11.67 ^b^	12.00 ^b^	13.35 ^a^	0.35	0.008	0.010	0.059	0.071
Leg °C	7:00	10.43 ^b^	11.15 ^ab^	11.92 ^a^	12.15 ^a^	0.46	0.059	0.013	0.310	0.602
	14:00	12.20 ^b^	13.13 ^ab^	13.33 ^ab^	14.20 ^a^	0.51	0.077	0.042	0.090	0.948
Rectal temperature °C	7:00	36.58	36.77	36.85	36.87	0.12	0.361	0.149	0.423	0.503
	14:00	37.60	37.78	37.58	37.72	0.07	0.189	0.043	0.576	0.737
Respiratory rate	7:00	10.77	11.33	11.17	11.22	0.28	0.533	0.290	0.623	0.373
	14:00	10.98 ^b^	11.17 ^b^	11.43 ^ab^	12.48 ^a^	0.39	0.055	0.127	0.277	0.034

^a, b^ Different superscripts within each row indicate significant differences (*p* < 0.05). The head, ear, abdomen, and leg represent the skin temperature of the site in °C, the rectal temperature in °C, and the respiration rate in times/min; WF: windproof facilities; WT: water temperature; WF × WT: the interaction between windproof facilities and water temperature.

**Table 4 animals-12-02405-t004:** Effects of housing and management systems on the blood biochemical indicators of donkeys in cold weather.

Items	Time	No Windproof f.		Windproof f.		SEM	*p*	*p*		
		Cold Water	Lukewarm Water	Cold Water	Lukewarm Water			WF	WT	WF × WT
21 d										
TP g/L	7:00	55.33 ^b^	58.17 ^b^	68.33 ^a^	71.75 ^a^	1.65	0.041	0.037	0.444	0.568
	14:00	55.33	59.83	61.33	62.67	2.22	0.155	0.053	0.432	0.177
ALB g/L	7:00	19.67	20.83	22.50	24.00	0.72	0.193	0.366	0.051	0.909
	14:00	19.83	20.33	21.17	23.00	1.13	0.830	0.414	0.632	0.784
TG mmol/L	7:00	0.15 ^b^	0.17 ^b^	0.21 ^a^	0.22 ^a^	0.01	0.046	0.033	0.647	0.844
	14:00	0.27	0.28	0.32	0.35	0.02	0.336	0.214	0.143	0.697
CHO mmol/L	7:00	1.52	1.57	1.58	1.58	0.05	0.973	0.714	0.789	0.814
	14:00	1.51	1.61	1.77	2.06	0.10	0.036	0.007	0.787	0.159
BUN mmol/L	7:00	5.44 ^b^	5.59 ^b^	6.21 ^a^	6.65 ^a^	0.22	0.043	0.012	0.855	0.944
	14:00	5.47	5.52	6.00	6.11	0.23	0.332	0.078	0.771	0.557
GLU mmol/L	7:00	7.05 ^a^	6.44 ^ab^	5.53 ^b^	5.23 ^b^	0.14	0.042	0.027	0.599	0.795
	14:00	4.49	4.72	4.90	4.98	0.32	0.278	0.062	0.824	0.518
LDL-C mmol/L	7:00	0.16	0.11	0.14	0.11	0.01	0.282	0.260	0.443	0.159
	14:00	0.12	0.11	0.15	0.17	0.01	0.143	0.032	0.795	0.424
HDL-C mmol/L	7:00	1.18 ^b^	1.19 ^b^	1.43 ^a^	1.49 ^a^	0.05	0.043	0.025	0.477	0.115
	14:00	1.53	1.52	1.62	1.66	0.04	0.479	0.126	0.957	0.889
42 d										
TP g/L	7:00	69.00 ^a^	67.50 ^a^	57.00 ^b^	56.67 ^b^	1.34	0.049	0.038	0.878	0.775
	14:00	59.33	58.50	56.50	56.00	1.76	0.830	0.852	0.458	0.609
ALB g/L	7:00	20.33	20.50	20.83	21.17	0.46	0.920	0.531	0.928	0.788
	14:00	20.50	20.67	20.17	21.17	0.66	0.959	0.950	0.661	0.754
TG mmol/L	7:00	0.19^b^	0.20 ^b^	0.29 ^a^	0.34 ^a^	0.01	0.038	0.034	0.813	0.348
	14:00	0.34	0.35	0.37	0.45	0.03	0.074	0.060	0.975	0.057
CHO mmol/L	7:00	1.40 ^b^	1.43 ^b^	1.60 ^a^	1.65 ^a^	0.06	0.043	0.037	0.272	0.882
	14:00	1.63	1.65	1.76	1.80	0.08	0.297	0.081	0.566	0.362
BUN mmol/L	7:00	5.52 ^a^	5.29 ^a^	4.71 ^b^	4.15 ^b^	0.14	0.045	0.028	0.319	0.355
	14:00	4.95	4.93	4.83	4.58	0.24	0.785	0.427	0.538	0.899
GLU mmol/L	7:00	6.02 ^a^	5.88 ^a^	5.11 ^b^	5.02^b^	0.10	0.047	0.022	0.777	0.671
	14:00	4.61	4.56	4.46	4.39	0.23	0.689	0.942	0.327	0.500
LDL-C mmol/L	7:00	0.07	0.06	0.11	0.11	0.01	0.174	0.031	0.764	0.819
	14:00	0.21	0.25	0.19	0.23	0.02	0.786	0.686	0.355	0.958
HDL-C mmol/L	7:00	1.23	1.27	1.38	1.35	0.05	0.099	0.059	0.836	0.776
	14:00	1.53	1.53	1.67	1.71	0.06	0.092	0.068	0.194	0.179

^a, b^ Different superscripts within each row indicate significant differences (*p* < 0.05). TP: total protein; ALB: albumin; TG: triglyceride; CHO: cholesterol; BUN: blood urea nitrogen; GLU: glucose; HDL-C: high-density lipoprotein cholesterol; LDL-C: low-density lipoprotein cholesterol; WF: windproof facilities; WT: water temperature; WF × WT: the interaction between windproof facilities and water temperature.

**Table 5 animals-12-02405-t005:** Effects of housing and management systems on the hormone levels of donkeys in cold weather.

Items	Time	No Windproof f.		Windproof f.		SEM	*p*	*p*		
		Cold Water	Lukewarm Water	Cold Water	Lukewarm Water			WF	WT	WF × WT
21 d										
ACTH pg/mL	7:00	64.22 ^a^	59.28 ^a^	53.97 ^b^	51.10 ^b^	1.81	0.045	0.034	0.693	0.319
	14:00	47.99	48.31	44.76	43.17	1.75	0.178	0.243	0.082	0.332
COR ng/mL	7:00	26.17 ^a^	25.23 ^a^	22.15 ^b^	21.99 ^b^	0.60	0.039	0.014	0.302	0.496
	14:00	20.43	19.77	19.99	19.15	0.45	0.727	0.331	0.798	0.678
ADPN μg/mL	7:00	9.26	9.66	9.10	10.08	0.14	0.141	0.660	0.034	0.345
	14:00	8.37	8.65	8.05	8.18	0.19	0.077	0.013	0.114	0.337
Insulin mIU/L	7:00	8.50	8.61	8.70	8.86	0.18	0.935	0.581	0.744	0.943
	14:00	8.17	8.35	8.31	8.76	0.18	0.709	0.505	0.748	0.452
Leptin ng/mL	7:00	2.67	2.65	2.86	2.82	0.07	0.651	0.222	0.851	0.931
	14:00	2.88	2.91	3.06	3.35	0.09	0.314	0.122	0.416	0.514
T3 nmol/L	7:00	2.25 ^a^	2.10 ^a^	1.83^b^	1.75 ^b^	0.04	0.036	0.022	0.863	0.492
	14:00	1.82	1.81	1.75	1.72	0.03	0.113	0.018	0.792	0.884
T4 nmol/L	7:00	43.51 ^a^	42.87 ^a^	38.95 ^b^	38.53 ^b^	0.93	0.043	0.028	0.832	0.779
	14:00	37.86	37.56	36.43	36.55	0.53	0.310	0.066	0.814	0.897
42 d										
ACTH pg/mL	7:00	50.89	50.16	48.80	47.59	0.95	0.480	0.703	0.247	0.325
	14:00	45.01	44.53	42.16	41.47	1.27	0.507	0.640	0.307	0.311
COR ng/mL	7:00	20.77	20.58	19.51	19.59	0.28	0.328	0.935	0.077	0.825
	14:00	17.74	17.67	17.35	17.06	0.41	0.739	0.460	0.994	0.413
ADPN μg/mL	7:00	8.53	7.91	8.85	8.01	0.18	0.254	0.584	0.067	0.764
	14:00	7.23	6.69	7.40	7.13	0.21	0.144	0.069	0.817	0.146
Insulin mIU/L	7:00	8.12	8.08	8.28	8.40	0.15	0.161	0.101	0.834	0.664
	14:00	8.08	8.02	8.12	8.15	0.19	0.617	0.829	0.329	0.389
Leptin ng/mL	7:00	2.10	1.90	2.08	1.93	0.06	0.524	0.996	0.150	0.820
	14:00	2.24	2.04	2.19	2.05	0.05	0.521	0.838	0.154	0.798
T3 nmol/L	7:00	1.59	1.49	1.55	1.51	0.03	0.766	0.918	0.363	0.637
	14:00	1.39	1.39	1.35	1.42	0.04	0.906	0.937	0.615	0.599
T4 nmol/L	7:00	40.77 ^a^	40.57 ^a^	36.58 ^b^	35.89 ^b^	0.88	0.046	0.026	0.678	0.599
	14:00	35.12	35.48	33.09	32.84	0.69	0.148	0.910	0.085	0.136

^a, b^ Different superscripts within each row indicate significant differences (*p* < 0.05). ACTH: adrenocorticotropic hormone; COR: cortisol; ADPN: adiponectin; INS: insulin; LEP: leptin; T3: triiodothyronine; T4: thyroxine; WF: windproof facilities; WT: water temperature; WF × WT: the interaction between windproof facilities and water temperature.

**Table 6 animals-12-02405-t006:** Effects of housing and management systems on the immune indexes of donkeys in cold weather.

Items	No Windproof f.		Windproof f.		SEM	*p*	*p*		
	Cold Water	Lukewarm Water	Cold Water	Lukewarm Water			WF	WT	WF × WT
21 d									
IL-1β pg/mL	128.6 ^a^	94.54 ^b^	80.46 ^c^	73.61 ^c^	2.38	<0.001	<0.001	0.001	0.012
IL-4 pg/mL	6.83 ^b^	6.96 ^b^	9.28 ^a^	10.69 ^a^	0.29	<0.001	<0.001	0.281	0.196
TNF-α pg/mL	26.32 ^a^	26.37 ^a^	18.61 ^b^	19.98 ^b^	0.74	0.001	<0.001	0.637	0.657
IgA μg/mL	35.54 ^b^	35.48 ^b^	51.30 ^a^	57.12 ^a^	1.83	0.001	<0.001	0.442	0.433
IgG μg/mL	43.64 ^b^	49.31 ^b^	69.76 ^a^	71.83 ^a^	1.73	<0.001	<0.001	0.278	0.610
IgM μg/mL	17.88 ^b^	18.94 ^b^	23.39 ^a^	25.37 ^a^	0.45	<0.001	<0.001	0.108	0.617
42 d									
IL-1β pg/mL	124.73 ^a^	115.69 ^a^	78.44 ^b^	76.89 ^b^	4.93	0.004	<0.001	0.004	0.006
IL-4 pg/mL	6.58 ^b^	7.65 ^b^	9.06 ^a^	10.47 ^a^	0.24	<0.001	<0.001	0.002	0.180
TNF-α pg/mL	29.09 ^a^	28.14 ^a^	19.97 ^b^	19.71 ^b^	0.88	0.010	<0.001	0.152	0.008
IgA μg/mL	37.80 ^b^	44.27 ^b^	54.38 ^a^	61.52 ^a^	2.62	0.009	0.005	0.220	0.951
IgG μg/mL	50.06 ^b^	55.18 ^b^	73.85 ^a^	79.91 ^a^	2.08	0.002	<0.001	0.206	0.914
IgM μg/mL	18.79 ^b^	20.94 ^b^	32.15 ^a^	32.37 ^a^	0.61	<0.001	<0.001	0.390	0.481

^a, b, c^ Different superscripts within each row indicate significant differences (*p* < 0.05). IL-1β: interleukin-1β; IL-4: interleukin-4; TNF-α: tumor necrosis factor-alpha; IgA: immunoglobulin A; IgG: immunoglobulin G; IgM: immunoglobulin M; WF: windproof facilities; WT: water temperature; WF × WT: the interaction between windproof facilities and water temperature.

**Table 7 animals-12-02405-t007:** Effects of housing and management systems on the antioxidative indexes of donkeys in cold weather.

Items	No Windproof f.		Windproof f.		SEM	*p*	*p*		
	Cold Water	Lukewarm Water	Cold Water	Lukewarm Water			WF	WT	WF × WT
21 d									
T-AOC mM	0.38	0.38	0.38	0.42	0.01	0.599	0.386	0.591	0.372
MDA nmol/mL	1.52 ^a^	1.44 ^a^	0.94 ^b^	0.84 ^b^	0.03	<0.001	<0.001	0.161	0.333
T-SOD U/mL	89.19 ^b^	91.16 ^b^	106.4 ^a^	108.0 ^a^	2.76	0.045	0.006	0.977	0.746
CAT U/mL	1.38	1.43	1.58	1.62	0.11	0.842	0.987	0.388	0.831
GPx U/mL	410.9 ^b^	487.3 ^b^	572.7 ^a^	613.2 ^a^	25.8	0.031	0.019	0.102	0.363
42 d									
T-AOC mM	0.39	0.37	0.40	0.38	0.01	0.754	0.749	0.312	0.979
MDA nmol/mL	1.52 ^a^	1.43 ^a^	1.00 ^b^	0.89 ^b^	0.09	0.009	0.005	0.288	0.243
T-SOD U/mL	88.26 ^b^	92.47 ^b^	118.0 ^a^	124.0 ^a^	3.08	0.010	<0.001	0.416	0.386
CAT U/mL	1.41 ^b^	1.55 ^b^	1.92 ^a^	2.06 ^a^	0.01	0.023	0.009	0.310	0.295
GPx U/mL	501.6	521.8	557.3	611.4	20.9	0.296	0.098	0.386	0.491

^a, b^ Different superscripts within each row indicate significant differences (*p* < 0.05). T-AOC: total antioxidant capacity; MDA: malondialdehyde; T-SOD: total superoxide dismutase; CAT: catalase; GPx: glutathione peroxidase; WF: windproof facilities; WT: water temperature; WF × WT: the interaction between windproof facilities and water temperature.

## Data Availability

The data that support the findings of this study are available from the corresponding authors upon reasonable request.

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
