# Peer review of "Effects of Housing and Management Systems on the Growth, Immunity, Antioxidation, and Related Physiological and Biochemical Indicators of Donkeys in Cold Weather"

_animals, 2022, doi:10.3390/ani12182405_

Round 1

Reviewer 1 Report

Thank you for submitting this manuscript. It considers the effect of housing on production parameters in donkeys. The study is interesting, but the results are not clearly stated, with some grammar and spelling mistakes that make sometimes the text difficult to read.

Please, delete the references, you only need to use numbers, not the name of the authors.

Where the parameters evaluated within normal range for the donkeys or were they abnormal at any point in time? Could you comment on that, please?

In the tables, please add the values at day 0. It is not clear, furthermore, if p is determined by the comparison between d0 and 21 or 42 or day 21 and 42. I would suggest revising the tables, to better explain the results of the statistical analysis.

Line 158: "collected in an empty stomach state at 07:00 and 14:00 on days 21 and 42", please specify. In the material and methods section, you wrote that hay was given at 13.00, was food withheld on days of blood sampling? Furthermore, gastric emptying is slow in donkeys, and empty stomach is achieve only after 18-24hrs of starving. I would suggest changing that sentence in "food was withheld for... before sampling"

Results: basic physiological indexes: it is not clear, both in the text and in table 3, the days you are comparing the indexes. You are only writing 07:00 and 14:00, but not the days you are analyzing (0? 21? 42?)

Lines 315-318: please add a reference to this sentence

Lines 381-388: in my opinion, your research shows only that better environmental conditions affect some of the hormones of the body, but you cannot prove an effect of central nervous system on the cold stress response.

Reviewer 2 Report

General comments: this is a well written study with a simple design that has important results for improving the welfare of donkeys.

In general your references seem to follow two formats, you have them written out and as numbers. Please check the journal formatting standards for the correct format.

Specific comments:

Line 14: “showed that adopting windproof”

Line 15: might use different wording than “indexes” such as “indicators”

Line 57: “have increased importance in meat and medicine production” might be a more clear phrasing

Line 58: remove “but” and replace with “world, the scientific approaches to their management”

Line 59: might give specific examples here. Unsure of what you mean by “cold-resistant animals”

Line 90: “animal houses, to allow animals to realize their productive potential” would be clearer

Line 168: “including the serum concentrations” instead of “contents”

Figure legends: make sure to make figure legends stand alone as descriptive of the figure. Such that readers would understand them without the paper. Also, describe whether reader is viewing mean ± SEM (from statistical output) or the average and SD.

Line 208: “intake of lukewarm water was greater than that of cold” no need to state significantly if you also have p values.

Line 227: here and throughout, remove the word “significantly”

Line 228: this is awkwardly phrased, maybe “when measured at 07:00, adopting windproof facilities increased respiration rates and skin temperatures, and tended to increase rectal temperatures.”

Table 3: it is redundant to say respiratory rate/minute. Rate infers the time. So would either say respiratory rate and then define “per minute” in a foot note or say “respirations per minute”

Line 321: when referring to storying certain energy, do you mean body fat? Also might consider the skin surface to body mass ratio of larger vs. smaller animals that also influences cold tolerance.

Line 363: what is meant by “imperfect development of various physiological features”? be more specific here.

Line 413: please add a reference for the brown fat function statement. Also, are you referring to all mammals here, or specific species?

Line 414: “main secretion of the thyroid gland”

Line 417: remove “significantly”

Reviewer 3 Report

This article provides a lot of quality data on the effects of cold stress on donkeys. There is limited research on the topic and this work will contribute positively to the literature. The experimental design was well thought out and the researchers analyzed a lot of parameters that provide strong evidence for their conclusions regarding the effectiveness of a windbreak on production traits and immune responses. The authors are encouraged to write more concisely and reword / rewrite a few sections for clarity. Tables need to be reformatted. 

Round 2

Reviewer 1 Report

Title: I would suggest changing "welfare management". It is not clear what you mean, please change the phrase to indicate what you mean for "welfare management": housing? watering? shelter? feeding systems?

Please check the manuscript for grammar and spelling mistakes, I still found some parts difficult to read. Furthermore, in the discussion section, the reading is made more difficult by the added sentences, please check for repetitions in the text.

Lines 57-59 "In spite of the great economic importance of donkeys in most countries of the world, the scientific approach to their management is generally ignored in the name of strong resistance animals, and donkeys' welfare is also often neglected." This sentence is not clear, please rephrase.

Lines 98-102: "welfare management" and "welfare facilities" are not suitable terms, it is not clear what you mean. I would suggest using different phrases, such as "different housing systems" or "housing and management systems that can improve the welfare of donkeys"

Material and methods, lines 109-119: please explain, it is still not clear. The donkeys without the windproof facilities were able to get outside the barn and those with were not? How was the lukewarm water delivered? And was it delivered to one barn with and one without the windproof facilities? If available, please add the measurement of the barns used.

Statistical analysis: in my opinion, you still need to compare the results of the evaluation at day 0 to determine if there were differences in the various parameters in the donkeys before the start of the experiment. This way you can then determine if what you found at day 21 and 42 was influenced by the different housing and water temperature

Lines 177-179: please add the number of the tables and figures

Figure 1, lines 193-197: please rephrase. Did you combine the temperature in the two barns with windproof facilities, the one with and the one without lukewarm water?

Results: there are a few papers and books indicating the reference range for various physiological and blood parameters in donkeys, for example, Goodrich and Behling-Kelly, Clinical Pathology of Donkeys and Mules, Vet Clin Equine 35 (2019) 433–455, presents a table with a summary of the relevant findings present in the literature. Please, compare these with your results. Are your parameters out of range at any time-point? I agree, that parameters of immune and antioxidant activity are less studied in donkeys, though.

Tables 2-7: please add the values of the parameters evaluated also at day 0, in my opinion, these results should still be included, even without the statistical analysis.

Lines 313-315: ulcers in stomach? please add "in rats"

Lines 316-336, please add references. You are writing "some studies", please add a refence to them. Also, when indicating that donkeys can store body fat and resist cold weather more efficiently, please add a reference

Lines 356-358: please add references

Round 3

Reviewer 1 Report

Thank you for addressing my concerns. I am sorry to say that, though, the fact that you did not collect the various parameters, except for body weight, at day 0 is, for me a flaw in the research. In my opinion, despite the young age and the uniformity of the donkeys enrolled, it is not possible to exclude disases in your population at day 0. Furthermore, for most of the parameters, you are comparing the two groups, without taking into account possible changes occuring over time also in the donkeys where the management was not changed. In my opinion, you cannot be sure that the animals, in the barn that was later protected from the wind and where lukewarm water was administered, were not already more comfortable than the other groups at the start of the study.

I have one more concern: were the windproof and non-windproof part of the barn separated by a barrier? 

Author Response

     We regret that can't provide data on day 0. Donkeys of similar weight were selected for the experiment and were checked by a vet to make sure they were healthy. Therefore, we believe that the body condition of donkeys was consistent before the experiment began. The space between the windproof and non-windproof barn is 10 meters, so the windproof facilities will not affect the environment of the non-windproof barn.